This is just a preview and not the published paper.



# Development and Assessment of a High Spatial Resolution (4.4 km) MISR Aerosol Product Using AERONET-DRAGON Data

**M. J. Garay[1], O. V. Kalashnikova[1], and M. A. Bull[1]**

[1]{Jet Propulsion Laboratory, California Institute of Technology, Pasadena, California}

Correspondence to: M. J. Garay (Michael.J.Garay@jpl.nasa.gov)

## Abstract

Since early 2000, the Multi-angle Imaging SpectroRadiometer (MISR) instrument on NASA's Terra satellite has been acquiring data that has been used to produce aerosol optical depth (AOD) and particle property retrievals at 17.6 km spatial resolution. Capitalizing on the capabilities provided by multiangle viewing, the current operational (Version 22) MISR algorithm performs well with about 75% of MISR AOD retrievals globally falling within 0.05 or 20% × AOD of paired validation data from the ground-based Aerosol Robotic Network (AERONET). This paper describes the development and assessment of a prototype version of a higher spatial resolution, 4.4 km MISR aerosol product compared against multiple AERONET Distributed Regional Aerosol Gridded Observations Network (DRAGON) deployments around the globe. We find that, overall, the 4.4 km AOD retrievals perform much better than the 17.6 km retrievals in comparison to the AERONET-DRAGON data for over 100 individual sites. In particular, the previously reported underestimation of AOD at high AOD in the operational 17.6 km product appears to be largely eliminated.

## 1   Introduction

Atmospheric aerosols, suspended particles of solid and liquid, play key roles in the weather and climate of the Earth. Aerosol optical depth (AOD) is a fundamental parameter that expresses the amount of aerosol in the atmospheric column and its effect on the transmission of sunlight. Global observations of aerosol amount depend fundamentally on retrievals of

This is just a preview and not the published paper.



AOD from instruments on satellite platforms, such as Multi-angle Imaging
SpectroRadiometer (MISR) and the MODerate resolution Imaging Spectroradiometer
(MODIS) that fly on the NASA Earth Observing System (EOS) Terra satellite. Satellite
aerosol observations are used to model the global radiation budget and investigate the effects
of aerosols on clouds (e.g., Boucher et al., 2013). Applications of satellite-derived AOD
information include air quality and health studies that use satellite-retrieved AOD to estimate
ground-level concentrations of particulate matter, especially particles with aerodynamic
diameter less than 2.5 μm ($PM_{2.5}$), which are known to have significant health effects due to
their ability to penetrate the human respiratory system (e.g., Martin, 2008; van Donkelaar et
al., 2015; 2016).
Critical to the success of satellite aerosol missions like MISR and MODIS are assessments of
the performance of their retrieval algorithms. Algorithm performance is typically evaluated
by the ability of the retrievals to capture the observed spatiotemporal variability of aerosols as
determined by ground-based observations, which are taken to represent the "truth." Within
the satellite aerosol community, the Aerosol Robotic Network (AERONET) is often used as a
standard, global reference. AERONET is a federated instrument network of ground-based
sunphotometers that derive AOD at a number of visible and near-infrared wavelengths from
direct sun observations (Holben et al., 1998).
The MISR instrument has been acquiring data from on board the NASA Terra Earth
Observing System (EOS) platform since early 2000. The current Level 2 (swath-based)
aerosol retrieval algorithm, designated F12_0022, or Version 22 (V22), began production at
the NASA Langley Research Center Atmospheric Science Data Center (ASDC) on 1
December 2007, and has been applied to the entire MISR mission, including current (forward)
processing. Details of the V22 MISR aerosol retrieval over water and land can be found in
Kalashnikova et al. (2013) and Martonchik et al. (2009), respectively. AOD and associated
aerosol particle properties are reported in the MISR aerosol product on a 17.6 km spatial
resolution grid, which represents 16 × 16 (256) samples of the 1.1 km resolution MISR
observations in four spectral bands in the visible and near infrared made from nine separate
viewing angles (Diner et al., 1998). The MISR aerosol product was evaluated against global
AERONET sites by Kahn et al. (2010), who reported that, overall, about 70% to 75% of
MISR AOD retrievals are within the greater of 0.05 or 0.2 × AOD of the paired AERONET
data. By way of comparison, the operational MODIS Collection 6 (C6) Dark Target (DT)

This is just a preview and not the published paper.



algorithm, which began production in 2014, has a reported expected error (EE) envelope,
containing about 67% of the retrievals relative to AERONET, of $-(0.02 + 0.1 \times AOD)$ to
$+(0.04 + 0.1 \times AOD)$ (Levy et al., 2013). Sayer et al. (2015) found that about 85% of
vegetated sites and 70% of arid sites fell within the EE envelope of $\pm(0.05 + 0.2 \times AOD)$ for
the MODIS C6 Deep Blue (DB) algorithm for MODIS-Terra after the application of
calibration corrections for the sensor.
Kahn et al. (2010) also identified a number of issues in the performance of the V22 MISR
aerosol retrieval algorithm, including a small gap at low AODs relative to AERONET and the
appearance of quantization noise, missing particles in the aerosol look up table, and a frequent
underestimate of AOD relative to AERONET over land when the AOD was greater than
about 0.4. Subsequently, Kalashnikova et al. (2013), Witek et al. (2013), and Shi et al. (2014)
identified issues with the cloud screening applied in the V22 algorithm, especially with regard
to thin cirrus, and suggested possible solutions; and Limbacher and Kahn (2015) diagnosed
the effects of stray light in the MISR cameras, noted earlier by Bruegge et al. (2002), that
could have significant impact on retrieved AODs in scenes with high contrast. These efforts
by members of the MISR science team and others have been directed at improving the quality
of the MISR aerosol product with the view of delivering a new version of the operational
MISR aerosol retrieval algorithm in the near future. At the same time, a number of studies
have highlighted the need for aerosol products at higher spatial resolutions than currently
available operationally from MISR and MODIS, gridded at 17.6 km and 10.0 km,
respectively. In response to this, the MODIS team released a global 3 km resolution DT
aerosol product as part of its Collection 6 delivery (Remer et al., 2013). In this work, we
describe the effort to develop a higher resolution, 4.4 km Level 2 MISR aerosol product based
on initial tests that showed significant retrieval improvement relative to AERONET sites
deployed in relatively large numbers in Distributed Regional Aerosol Gridded Observations
Network (DRAGON) campaigns around the globe (e.g., Eck et al., 2014; Seo et al., 2015;
Sano et al., 2016). In the DRAGON networks, instruments are located much closer to one
another, with a typical grid spacing around 10 km (e.g., Munchak et al., 2013). As we will
discuss in this paper, we found that the new MISR 4.4 aerosol product is better able to resolve
spatial gradients in AOD compared to the operational (V22) 17.6 km resolution product as
shown in a number of comparisons from different DRAGON deployments that encompass a
wide range of aerosol loadings.

This is just a preview and not the published paper.





## 2 Data and methods

### 2.1 20 January 2013 MISR overpass of DRAGON San Joaquin Valley, California

The initial motivation for this work was a MISR overpass of the DRAGON sites deployed in the San Joaquin Valley of California in support of the NASA Deriving Information on Surface conditions from Column and Vertically Resolved Observations Relevant to Air Quality (DISCOVER-AQ) field campaign in January and February 2013 (see Beyersdorf et al., 2016). Figure 1a shows the red band (672 nm) image from MISR Orbit 69644 when the Terra satellite passed over the San Joaquin Valley around 18:50 UTC on 20 January 2013. The image is oriented with north to the top. The bright features in the upper central portion of the image are snow in the Sierra Nevada, with the San Joaquin Valley of central California to the southwest. Figure 1b shows the green band (558 nm) AOD reported in the MISR V22 operational aerosol product at 17.6 km resolution. The circles correspond to the AODs reported by the AERONET-DRAGON sites closest in time to the Terra overpass using the same color scale as the MISR AODs. The horizontal lines denote the MISR "blocks" that correspond to 141 km in the along-track direction of the satellite motion (Bothwell et al., 2002). It is clear in Fig. 1b that the aerosols are concentrated in the San Joaquin Valley, although on this date the AOD is relatively low, with a maximum around 0.30.

As mentioned above, the V22 MISR aerosol retrieval algorithm takes as input the 256 – 1.1 km MISR Level 1B2 pixels within the 17.6 km retrieval region (16 pixels × 16 pixels). In standard "global" acquisition mode, blue, green, and near infrared bands in the off-nadir cameras are averaged onboard from the full 275 m pixel resolution to 1.1 km to save data rate, while the red bands in all nine cameras and the blue, green, and near infrared bands for the nadir camera are preserved at their full resolution (Diner et al., 1998). The 1.1 km pixel data for the red band and the nadir camera are calculated by the aerosol algorithm by simple averaging. The MISR instrument has another "local" acquisition mode that preserves the full resolution of the data for all nine cameras and four spectral bands for a target with an along-track length of about 300 km (Diner et al., 1998). It was recognized that with some modifications the V22 algorithm could be applied to this "local mode" data, resulting in a product with 4.4 km spatial resolution due to the change of the input resolution from 1.1 km

This is just a preview and not the published paper.





to 275 m (since 275 m × 16 = 4.4 km). Figure 1c shows the results of the application of the
modified V22 algorithm to a local mode acquisition made over Pixley, CA (PIXLEYCA),
which accounts for the smaller geographic coverage of the retrieval. The same color scale is
applied to the AOD retrievals in this case as in Fig. 1b, and, again, the AERONET-DRAGON
sites are indicated by circles colored by the AOD reported for the time nearest the Terra
overpass. Not only is much greater detail revealed regarding the spatial distribution of
aerosols in the San Joaquin Valley, with higher aerosol loading extending from the Fresno in
the central part of the valley to Bakersfield in the southeast, but visually the agreement
between the MISR AODs and the AERONET-DRAGON AODs is much improved.
The visual impression of better agreement is borne out in the regression analysis shown in
Fig. 2. Figure 2a compares the 17.6 km V22 AODs from MISR at 558 nm with the
AERONET AODs linearly interpolated from the two nearest wavelengths on either side in
log-log space to 558 nm (e.g., Sayer et al., 2013). The matches are made nearest in time to
the Terra overpass (typically within 15 minutes) and the AERONET observations are required
to fall within a specific 17.6 km retrieval region. These criteria are somewhat different than
the matching criteria used in Kahn et al. (2010), who considered the average AOD of
AERONET observations within a 2 h window centered at the time of the satellite overpass,
with at least one valid observation within the hour before and one in the hour after. They also
considered MISR retrievals in both the "central" 17.6 km region and the eight surrounding
regions. The interpolation of the AERONET AODs to the MISR wavelength was also done
slightly differently by Kahn et al. (2010), who used a second order polynomial fit, but this
resulted in a negligible change in the results for this particular case. As in Kahn et al. (2010)
the analysis here uses the "best-estimate" MISR AODs, which correspond to the mean of the
AODs for all the mixtures in the MISR look up table that pass the acceptance criteria. For the
17.6 km MISR retrieval there are 11 temporal and spatial matches with the AERONET data.
The correlation coefficient, $r$, is 0.6563; the root mean squared error (RMSE) is 0.0499; the
bias is −0.0233; and the percent of data within the EE envelope of MISR (the greater of 0.05
or 0.20 × AOD) is 72.73%. These results show the 17.6 km V22 retrieval performs relative to
the AERONET-DRAGON observations in a way that is generally consistent with the global
performance of the algorithm as assessed by Kahn et al. (2010).
Figure 2b shows the regression of the 4.4 km MISR aerosol retrieval using the slightly
modified V22 algorithm and the 275 m local mode input. Now, the correlation coefficient is

This is just a preview and not the published paper.





0.9144, the RMSE is 0.0184, the bias is –0.0060, and 100% of the data fall within the EE
envelope. These are all significant improvements in the agreement between the MISR aerosol
retrieval and the AERONET-DRAGON observations. The sampling was reduced by two
points, but inspection of the results shows that the data points that were eliminated due to the
requirement that the AERONET site fall within the 4.4 km retrieval region were both already
in good agreement with the 17.6 km MISR aerosol retrieval, which means that the
improvement is not simply due to the exclusion of outliers in the comparison. Over years of
refinements applied to the 17.6 km retrieval to improve performance relative to AERONET,
the results in Fig. 2 are among the most significant that were ever obtained. Note that these
results are also in contrast to the results of Remer et al. (2013) regarding the MODIS 3 km DT
retrievals. They reported that agreement of the 3 km retrieval relative to AERONET was
slightly worse over land compared to the 10 km retrieval, while the performance was similar
over ocean. The EE envelopes were found to be ±0.05 ±0.20 × AOD and ±0.03 ±0.05 × AOD
for land and ocean, respectively.
A further point is that the unique, high density nature of the AERONET-DRAGON
deployment is important for assessing the ability of a high resolution aerosol retrieval
algorithm to capture the true spatial variability of aerosols within a region. As shown in Fig.
1, the higher resolution MISR aerosol retrieval is better able to represent the spatial gradients
in the aerosol load, even though the aerosol load is relatively low on this date and aerosols are
spread throughout the San Joaquin Valley. In this case, both the 17.6 km and 4.4 km
retrievals report nearly identical values for the Fresno_2 site AERONET site, which is the
"permanent" site in the San Joaquin Valley and not part of the DRAGON deployment. So
comparisons with this single site would not reveal any important difference in the two
versions of the algorithm. Of course, a single case cannot support the conclusion that the 4.4
km MISR retrieval is superior to the 17.6 km retrieval in an overall sense, so further
comparisons were made with AERONET-DRAGON deployments around the globe in a
variety of aerosol loading situations. Even so, the results from the 20 January 2013 case were
sufficiently encouraging to focus the MISR science team on the development of a 4.4 km
spatial resolution retrieval that would not rely on local mode data to achieve the resolution
improvement, but would work with the 1.1 km global mode data.

This is just a preview and not the published paper.



## 2.2 AERONET-DRAGON deployments
According to the AERONET website (http://aeronet.gsfc.nasa.gov/new_web/dragon.html),
there have been nine AERONET-DRAGON deployments between 2011 and 2016. However,
the 20 January 2013 MISR case was instructive in terms of specific characteristics of a
deployment necessary to facilitate a comparison of the 17.6 km resolution aerosol retrieval
with a higher resolution 4.4 km retrieval. The primary consideration involves the number and
density of sites in the deployment. Table 1 shows an evaluation of eight of the nine
DRAGON deployments in terms of the spatial statistics. The on-going deployment of
DRAGON as part of the KORUS-AQ field campaign in South Korea, Japan, and China was
not considered here.
Starting with the San Joaquin Valley deployment, the table shows that 28 sites were deployed.
This results in 378 pairs (28 choose 2). Calculating the separation between each pair, there
are seven pairs separated by less than 17.6 km, 3 pairs separated by less than 8.8 km, and 1
pair separated by less than 4.4 km. The mean distance between pairs is 245.7, while the
median distance is 204.8 km. The MISR analysis is facilitated by a relatively large number of
pairs separated by less than 17.6 km that can be used to test the ability of the 4.4 km
algorithm to retrieve spatial gradients, but few pairs separated by less than 4.4 km, which will
likely fall inside a single 4.4 km retrieval region. The swath and orbit characteristics of MISR
must also be taken into account. MISR has a swath of about 400 km and Terra has a repeat
cycle of 16 days. Deployments with widely separated clusters of sites will therefore only
provide a limited number of comparisons for a particular MISR overpass. Cloudiness is a
further consideration as the DRAGON deployments typically happen within a limited time
frame – about a month in the case of San Joaquin Valley.
Based on these considerations, and visual inspections of candidate scenes, a set of MISR
cases was identified during DRAGON deployments for testing the 4.4 km and 17.6 km
resolution aerosol retrievals relative to AERONET. This set is shown in Table 2. In the table,
the "SOM Path" corresponds to the Space-Oblique Mercator (SOM) projection onto the
World Geodetic System 1984 (WGS84) ellipsoid used for the MISR processing (Diner et al.,
1998). There are 233 SOM paths within each 16-day repeat cycle of Terra. The cases are
broadly classified in terms of the range of AODs, with "low AOD" representing AODs
generally less than 0.3, "moderate AOD" corresponding to AODs between about 0.3 and 0.6,
and "high AOD" having AODs between about 0.6 and 1.4. Note that while the cases are

This is just a preview and not the published paper.





distributed globally including Washington D.C./Baltimore; the San Joaquin Valley in
California; Seoul, South Korea; and Osaka, Japan; a limitation of this study is that the
AERONET-DRAGON deployments have been primarily to mid-latitude locations, so there
are no cases from tropical, arid desert, or polar regions.
Figure 3 provides maps of the four relevant AERONET-DRAGON deployments. Figure 3a
shows the locations of 45 of the 46 sites deployed in 2011 for the Washington,
D.C./Baltimore campaign. The sites are generally located around the greater Baltimore area.
For reference, the distance between Washington, D.C. and Baltimore, MD is about 56 km.
Also, recall that 1 degree of latitude corresponds to about 111 km, which provides another
reference. Figure 3b shows the 25 sites deployed in South Korea in 2012. The majority of
sites are clustered around Seoul with a relatively large number of sites spaced less than 4.4
km apart, as shown in Table 1. Even so, the overall number of sites makes this a reasonable
test case for the 4.4 km MISR algorithm. Additionally, the cases from South Korea offer
situations with high AOD. Figure 3c shows the 18 AERONET-DRAGON sites deployed in
the San Joaquin Valley of California. Compared to the other cases, the density of sites in this
deployment is somewhat less, but this provides good sampling of the aerosol distribution
throughout the valley. Finally, Fig. 3d shows the locations of the 14 AERONET-DRAGON
sites deployed around Osaka, Japan in 2012. The largest density of sites is around Osaka,
itself. Again, the spatial clustering of sites is less than ideal, since many of them are separated
by less than 4.4 km.
**2.3  MISR aerosol retrievals over land**
Details of the MISR aerosol retrieval over land, which is most relevant to comparisons with
AERONET-DRAGON, can be found in Martonchik et al. (2009). The fundamental principal
of the retrieval is the separation of the multi-angular satellite signal at the top of the
atmosphere (TOA) into a component due to the aerosols and a component due to multiple
surface-atmosphere interactions. The primary underlying physical assumptions are the
following:
1. Aerosols are horizontally homogeneous in the retrieval region.
2. A predefined set of aerosols stored in a look up table is applied globally to retrievals
over both land and water.

This is just a preview and not the published paper.





3.  One or more cost functions ($\chi^2$ parameters) are assessed to determine how well

2       modelled TOA radiances from individual aerosol models and associated green-band

3       AODs match the observed TOA radiances.

4.  The angular shape of the surface reflectance is assumed to be spectrally invariant and

5       this is used to filter out models and AODs that do not conform to this assumption as

6       being unlikely candidates for selection (Diner et al., 2005).

5.  There is sufficient surface contrast in the retrieval region so that the TOA radiances

8       can be represented by empirical orthogonal functions (EOFs) generated directly from

9       the multiangle imagery.

6.  No retrievals are performed over complex terrain.
The choice of acceptable 1.1 km subregions within the retrieval region is done through the
application of a number of tests including cloud masking.

## 14   3    Results

### 15   3.1   Regression analysis

Figure 4a shows the regression of the V22 17.6 km MISR green-band AODs against the
AERONET-DRAGON AODs interpolated to the MISR wavelength for all the cases listed in
Table 2.  The range of AODs in this figure is much greater than the AOD range in Fig. 2.
Like the regressions shown in Kahn et al. (2010) the underestimation of the retrieved AODs
relative to AERONET for AODs greater than about 0.4 is apparent in this figure.  By way of
comparison, Fig. 4b shows the regression for a prototype 4.4 km MISR aerosol retrieval
(internally designated V22b24-34+1) that takes the 1.1 km spatial resolution global mode data
as input.  Tests showed that the retrieval algorithm did not perform significantly differently
when using 275 m local mode data as input compared to the 1.1 km global mode data.
The primary difference apparent in Fig. 4 is the improved performance of the 4.4 km
algorithm at high AODs.  The fall-off evident in the V22 17.6 km resolution retrievals is
greatly mitigated, if not eliminated entirely, but it is difficult to tell if any residual bias exists
at large AODs due to the small sample size in this AOD range.  Comparing the statistics, the
sampling is much greater for the 4.4 km resolution retrieval.  This is primarily due to the
relaxation of the thresholds on the $\chi^2$ parameters to admit better spatial coverage in the 4.4 km

This is just a preview and not the published paper.





retrieval. The need for this change was apparent when looking at maps constructed from the
4.4 km retrievals using the initial thresholds. The other parameters all show significant
improvements as well. The correlation coefficient goes from 0.8772 to 0.9595; the RMSE
decreases from 0.1683 to 0.0768; the bias decreases, in an absolute sense, from –0.0887 to –
0.0208, driven primarily by the improvement in the performance of the algorithm at large
AODs; and the percent of data within the MISR EE envelope increases from 59.09% to
80.92%. Although the statistics from this sample are insufficient for a complete analysis, the
last result suggests that the performance of the 4.4 km resolution algorithm will permit the
setting of a somewhat tighter EE envelope.
**3.2   Example images**
Besides providing improved regressions in AOD when compared with observations from the
AERONET-DRAGON sites, the greatest benefit of the 4.4 km resolution MISR aerosol
retrievals is most apparent when comparing maps of the retrieved AOD with the operational
V22 17.6 km algorithm. Figure 5 shows the MISR AOD retrievals for Orbit 65731 over
Osaka, Japan on 27 April 2012 at about 01:55 UTC. As shown in the MISR red band image
in Fig. 5a, the scene is extremely clear. The retrieved AODs on this day range up to about 0.3
in the vicinity of Osaka, itself. The main difference between the V22 17.6 km AOD map in
Fig. 5b and the 4.4 km retrieval map in Fig. 5c is the improvement in coverage due to the
relaxation of the $\chi^2$ thresholds in the 4.4 km retrieval. The remaining missing retrievals,
indicated in white, are due primarily to the shallow water between Honshu, the main landmass
in the upper (northern) portion of the image, and the mountainous island of Shikoku. The
MISR Dark Water algorithm does not attempt to perform retrievals in locations identified as
"shallow water" due to possible contributions from the underwater surface (e.g., Kahn et al.,
2009). Retrievals are also not performed over much of Shikoku due to the complex terrain in
the mountains, which violates the assumptions of the 1-D radiative transfer used in the MISR
aerosol retrieval algorithm. Although these exclusion conditions apply to both the 17.6 km
and 4.4 km algorithms, the higher resolution retrieval typically obtains better coverage by
being able to get closer to these exclusion zones. Some of the improved coverage of the 17.6
km retrieval, in the lower right portion of the image, for example, is only due to the larger
area covered by a single 17.6 km pixel, compared to a single 4.4 km pixel.

This is just a preview and not the published paper.





Figures 6 and 7 show the spatial sampling over South Korea for cases with very high aerosol
loads. The white regions in the MISR red band image in Fig. 6a are clouds to the northeast
and southwest of the peninsula on 9 May 2012 at the Terra overpass time around 02:20 UTC.
The landmass, however, is mainly clear. The V22 17.6 km retrieval does not have coverage
over most of the region, and the agreement between the MISR AODs and the AERONET-
DRAGON sites (colored circles) is not particularly good. The latter is not surprising given
the underestimation at high AOD in the 17.6 km product apparent in Fig. 4a. The 4.4 km
aerosol retrieval in Fig. 6b has much better coverage, with the missing locations
corresponding well with areas with large amounts of topographic relief. What is particularly
striking is the ability of this retrieval to capture the true spatial variability of the aerosol
throughout the region, in good agreement with the AERONET-DRAGON observations. In
this case, there does not appear to be any high bias due to the presence of urban surfaces,
which has been identified as an issue in the MODIS 3 km aerosol product (Munchak et al.,
2013). Unfortunately, without ancillary information it is difficult to assess the veracity of the
high AODs shown in the vicinity of the clouds in the far right of the image. However, both
the 17.6 km and 4.4 km retrievals indicate elevated AODs in this area.
The case in Fig. 7 has somewhat lower AODs than the previous case. Figure 7a shows the
MISR red band image from 25 May 2012 at around 02:20 UTC. There are orographic clouds
along the eastern coast of the Korean peninsula and a solid mass of clouds in the lower right
of the image. Again, the V22 17.6 km resolution product shown in Fig. 7b has missing
retrievals over much of the landmass. However, there appears to be a northwest to southeast
gradient in the AODs, continuing over the water. Figure 7c shows that evidence for this
overall gradient is lacking by filling in many of the missing areas. Instead, locations of high
AOD appear sporadically in the scene. The highest AODs are found over Seoul, which has
the majority of the AERONET-DRAGON sites, a couple of locations to the southeast, and
near the edges of the cloud fields. The two locations to the southeast of Seoul correspond to
valleys that are likely trapping pollution on this particular date. Again, it is hard to assess the
veracity of the high AODs in the lower portion of the image, but at least the two algorithms
are consistent with one another.

This is just a preview and not the published paper.



## 4   Discussion and conclusions

The operational V22 MISR aerosol retrieval algorithm went into production in December 2007. Since that time other satellite aerosol retrieval products have undergone significant enhancements, including both the MODIS DT and DB algorithms (Levy et al., 2013; Remer et al., 2013; Sayer et al., 2015). Efforts to improve the MISR aerosol algorithm have focused on the issues noted by Kahn et al. (2010) in their evaluation of the MISR V22 aerosol product against global AERONET observations, as well as topics raised by others (e.g., Kalashnikova et al., 2011; Witek et al., 2013; Shi et al., 2014; and Limbacher and Kahn, 2015). While the air quality community has raised the issue of spatial resolution in terms of using satellite data to study the health impacts of atmospheric aerosols on the appropriate "neighborhood scales," on the order of one or a few kilometers, the biggest surprise in moving to a higher resolution was the improvement in the retrieved AOD relative to AERONET – an improvement that did not require significant changes to the algorithm itself. This was surprising for two reasons. First, the more or less accepted line of thinking is that aerosols are generally spatially homogeneous at scales of 10's to 100's of kilometers, and temporally stationary, in a statistical sense, at time scales of hours to days (e.g., Anderson et al., 2003). Secondly, the MODIS team did not find significant improvement in the performance of their algorithm when they increased the resolution from 10 km to 3 km (Remer et al., 2013). In fact, this change in resolution highlighted some underlying issues in the assumptions going into the DT retrieval (Munchak et al., 2013).

Importantly, it would have been difficult to assess the performance of a high-resolution algorithm without appropriate high-resolution observations to evaluate against. A single AERONET site basically returns a "point" in space and time relative to retrievals from a satellite instrument. This has led to the adoption of averaging approaches that require large amounts of paired satellite-AERONET data matched within relatively broad spatial and temporal windows (e.g., Ichoku et al., 2003; Kahn et al., 2010; Petrenko et al., 2012). The deployment of AERONET-DRAGON sites beginning in 2011 has been a game-changer in terms of the ability to truly consider aerosol spatial variability, and the DRAGON deployments at sites around the globe facilitated the analysis presented here.

The performance of the operational V22 17.6 km MISR aerosol retrieval relative to the performance of a prototype 4.4 km retrieval was assessed in comparisons with multiple AERONET-DRAGON deployments over a broad range of AODs. It was found that, overall,





the 4.4 km retrieval performed significantly better than the 17.6 km retrieval. Part of the
reason for this improvement is the ability of the higher-resolution retrieval to capture the true
spatial variability of the aerosols, which is also captured by the DRAGON network. Again, a
single AERONET site cannot directly represent the spatial variability of aerosols, although
this is aliases into the temporal dependence of the AODs reported by the instrument.
Averaging the AERONET data over a time window and the satellite data over a spatial
window, as is traditionally done in global comparisons, has the effect of minimizing the
contributions of true aerosol spatial variability. Another reason for the improvement of the
MISR retrieval algorithm when applied at 4.4 km is that the assumptions underlying the
aerosol retrieval, particularly over land, are better met at this higher spatial resolution.
Ironically, among the most critical of these assumptions is that aerosols are spatially
homogeneous on the scale of the retrieval. In other words, aerosol variability itself is one of
the issues with the 17.6 km retrieval.
The MISR aerosol algorithm team is working toward the release of an updated version of the
aerosol retrieval that will have results reported globally at 4.4 km resolution. In addition to
this change, other changes and being tested and implemented with regard to cloud screening,
per-retrieval uncertainty reporting, and microphysical property retrievals. Critical to the
development of this new algorithm are assessments against AERONET-DRAGON
deployments for a range of cases represented by those used in this paper.
**Acknowledgements**
This work was performed at the Jet Propulsion Laboratory, California Institute of Technology
under a contract with the National Aeronautics and Space Administration. The MISR data
used in this work were obtained from the NASA Langley Research Center Atmospheric
Science Data Center. We thank the many PI investigators, and particularly the hard work of
Brent Holben and his team for establishing and maintaining the AERONET and AERONET-
DRAGON sites used in this investigation.

This is just a preview and not the published paper.



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

This is just a preview and not the published paper.



1     Table 1. Spatial statistics of AERONET-DRAGON deployments.

| DRAGON Campaign | Sites | Pairs | Separation < 17.6 km | Separation < 8.8 km | Separation < 4.4 km | Mean Separation (km) | Median Separation (km) |
|---|---|---|---|---|---|---|---|
| USA 2011 (Washington D.C., Baltimore) | 46 | 1035 | 105 | 21 | 2 | 51.4 | 42.6 |
| Asia 2012 (Japan, South Korea) | 53 | 1378 | 54 | 22 | 11 | 525.9 | 543.0 |
| SE Asia 2012 (7-SEAS) | 46 | 1035 | 31 | 8 | 3 | 1927.0 | 1877.5 |
| USA 2012-2013 (San Joaquin Valley) | 28 | 378 | 7 | 3 | 1 | 245.7 | 204.8 |
| Germany 2013 (HOPE) | 15 | 105 | 3 | 3 | 1 | 359.2 | 397.4 |
| USA 2013 (Houston) | 19 | 171 | 6 | 2 | 1 | 103.3 | 66.1 |
| USA 2013 (SEAC[4]RS) | 54 | 1431 | 9 | 5 | 3 | 993.6 | 989.0 |
| USA 2014 (Colorado) | 15 | 105 | 6 | 1 | 0 | 87.1 | 53.1 |

This is just a preview and not the published paper.





Table 2. MISR cases for AERONET-DRAGON comparison.

| Orbit | Date/Time | Campaign | SOM Path | MISR Blocks | Notes |
|---|---|---|---|---|---|
| 60934 | 2011-06-02 16:05 UTC | Washington, Baltimore | 16 | 58-60 | Low AOD, Clear |
| 61633 | 2011-07-20 16:05 UTC | Washington, Baltimore | 16 | 58-60 | Moderate AOD, Scattered Clouds |
| 61662 | 2011-07-22 15:55 UTC | Washington, Baltimore | 14 | 58-60 | Moderate AOD, Scattered Clouds |
| 65440 | 2012-04-07 02:20 UTC | Asia-Seoul | 115 | 60-62 | Low AOD, Clear |
| 65731 | 2012-04-27 01:55 UTC | Asia-Osaka | 111 | 62-64 | Low AOD, Clear |
| 65775 | 2012-04-30 02:25 UTC | Asia-Seoul | 116 | 60-62 | Moderate AOD, Clear |
| 65906 | 2012-05-09 02:20 UTC | Asia-Seoul | 115 | 60-62 | High AOD, Hazy |
| 66139 | 2012-05-25 02:20 UTC | Asia-Seoul | 115 | 60-62 | High AOD, Hazy |
| 69644 | 2013-01-20 18:50 UTC | San Joaquin Valley | 42 | 60-63 | Low AOD, Clear |
| 69877 | 2013-02-05 18:50 UTC | San Joaquin Valley | 42 | 60-63 | Moderate AOD, Few Clouds |

This is just a preview and not the published paper.





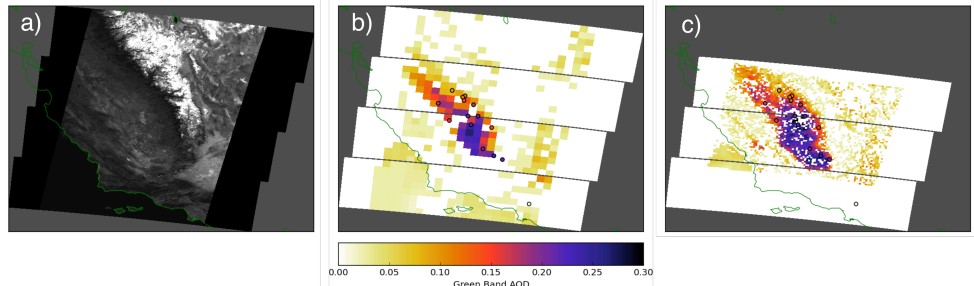

Figure 1. (a) MISR red band image of the San Joaquin Valley in California on 20 January
2013 at around 18:50 UTC; (b) MISR V22 17.6 km aerosol optical depth (AOD); (c) MISR
4.4 km AOD retrieved using a modified version of the V22 aerosol retrieval algorithm with
275 m local mode data as input.

This is just a preview and not the published paper.




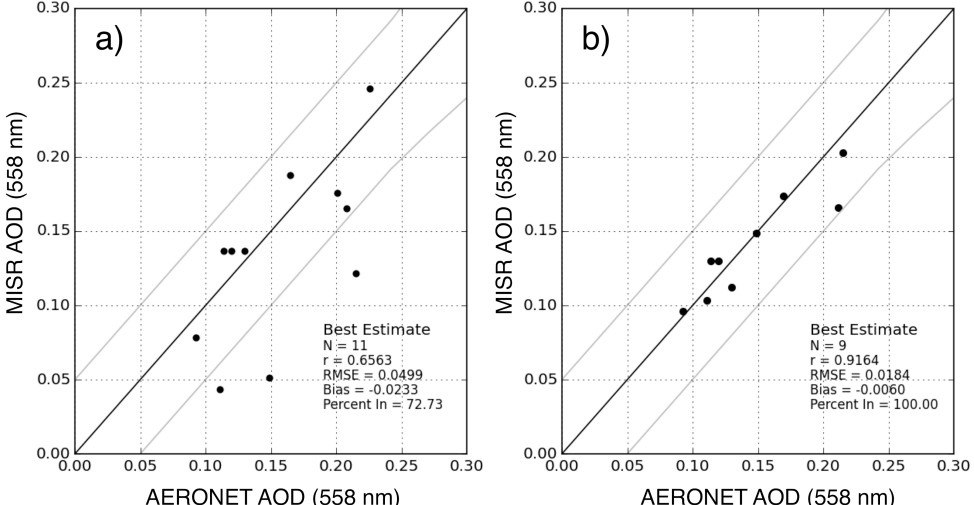

2    Figure 2. (a) Regression of MISR V22 17.6 km resolution AODs against AERONET

3    interpolated to the MISR wavelength. (b) Regression of the 4.4 km resolution AODs against

4    AERONET.

This is just a preview and not the published paper.



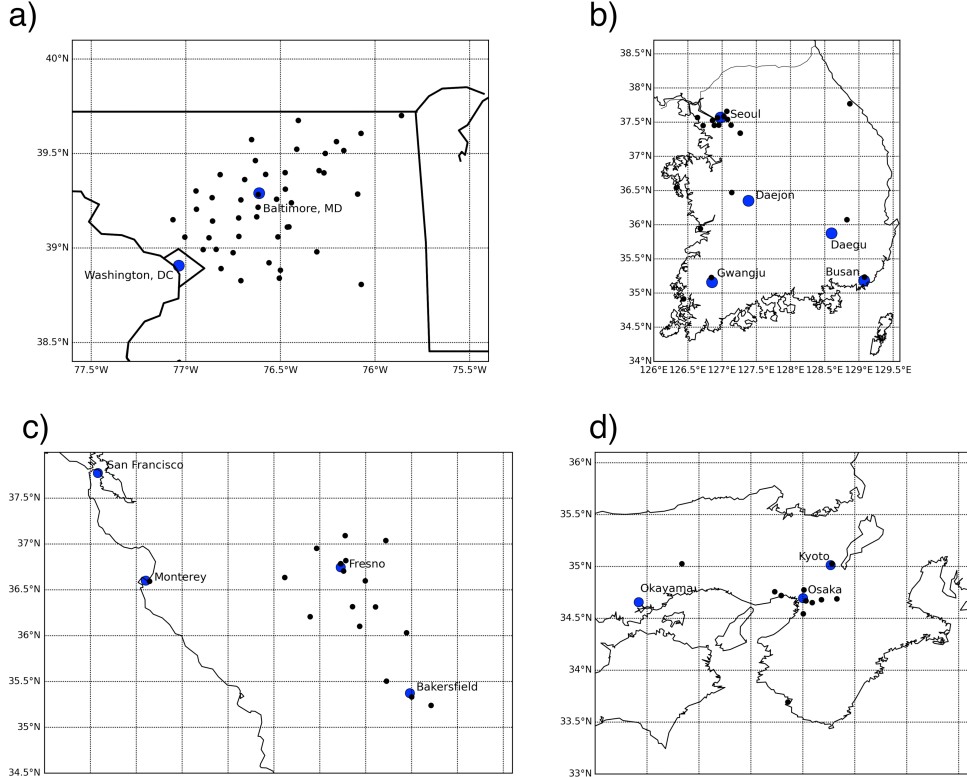

Figure 3. (a) Locations of the 45 sites deployed as part of the AERONET-DRAGON campaign in Washington, D.C./Baltimore metropolitan area; (b) Locations of the 25 sites deployed in South Korea during DRAGON-Asia 2012; (c) Locations of the 18 sites deployed in the San Joaquin Valley in California in late 2012 and early 2013; (d) Location of the 14 sites deployed in Japan during DRAGON-Asia 2012.

This is just a preview and not the published paper.





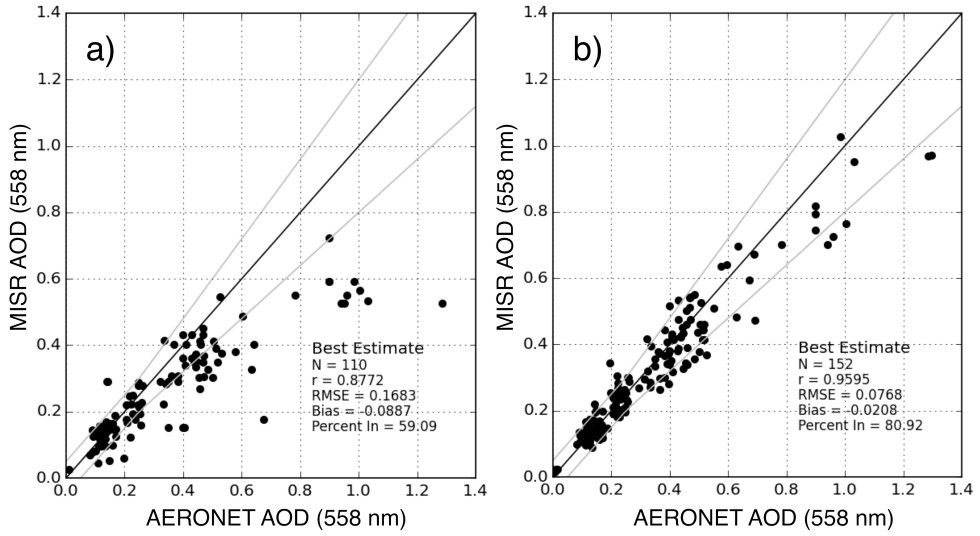

2    Figure 4. (a) Regression of MISR V22 17.6 km resolution AODs against AERONET

3    interpolated to the MISR wavelength. (b) Regression of the 4.4 km resolution AODs against

4    AERONET.

This is just a preview and not the published paper.





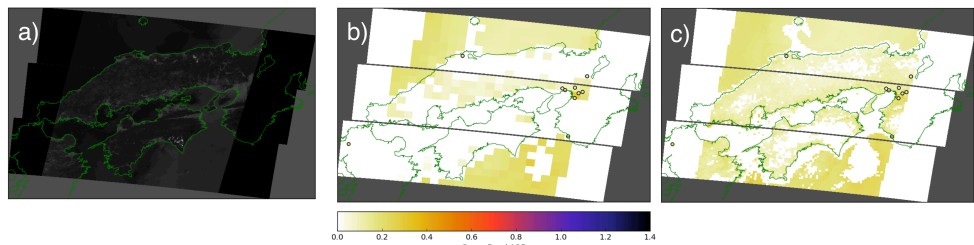

2   Figure 5. (a) MISR red band image of the Osaka, Japan on 27 April 2012 at around 01:55

3   UTC; (b) MISR V22 17.6 km aerosol optical depth (AOD); (c) MISR 4.4 km AOD retrieved

4   using a prototype algorithm that takes the 1.1 km global data as input.

This is just a preview and not the published paper.





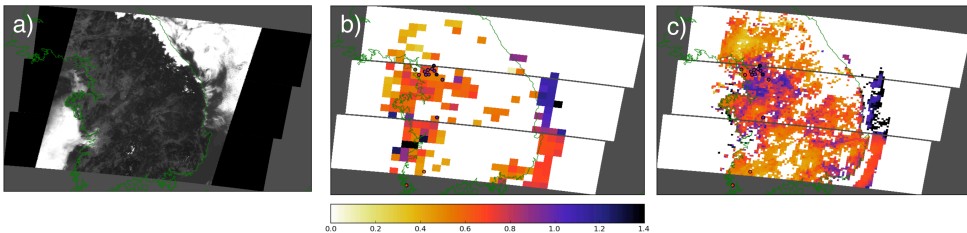

2 Figure 6. (a) MISR red band image of the Korea on 09 May 2012 at around 02:20 UTC; (b)

3 MISR V22 17.6 km aerosol optical depth (AOD); (c) MISR 4.4 km AOD retrieved using the

4 prototype algorithm.

This is just a preview and not the published paper.



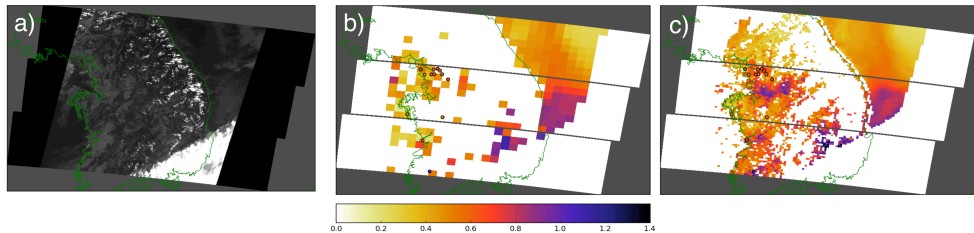

2    Figure 7. (a) MISR red band image of the Korea on 25 May 2012 at around 02:20 UTC; (b)

3    MISR V22 17.6 km aerosol optical depth (AOD); (c) MISR 4.4 km AOD retrieved using the

4    prototype algorithm.