# Peer review of "Development and Assessment of a Higher Spatial Resolution (4.4 km) MISR Aerosol Optical Depth Product Using AERONET-DRAGON Data"

_Atmospheric Chemistry and Physics, 2016_

## Referee Comment (RC1) · A. M. Sayer (Referee) · 16 Aug 2016

This paper illustrates how the MISR standard aerosol retrieval algorithm's performance improves, compared to AERONET DRAGON deployments (Sun photometers deployed in distributed networks for field campaigns), when the output horizontal pixel size of the algorithm is reduced from 17.6 km to 4.4 km. This increased spatial resolution is one of the changes which will be featured in the next version of the MISR standard aerosol product.

I was initially surprised that this manuscript was in ACPD rather than AMTD, since it is related to remote sensing algorithm updates. However, since the paper is mainly an illustration of results rather than an algorithm description or theoretical analysis, Printer-friendly version

I think it can fit in ACPD as well. The quality of language is good and I don't have any major issues with the manuscript. There are however some topics which aren't discussed within the manuscript, which I think shouldn't be too difficult to add, and would increase the interest/usefulness without making the manuscript overly long. I therefore favour minor revisions, and would be willing to review the revised manuscript if necessary.

General comments:

1. As the authors note, the MODIS 3 km aerosol product was found to have poorer performance than the nominal 10 km product when compared to AERONET, which is the converse of the authors' experience with MISR, where the higher resolution improves things. My understanding is that with MODIS this is mostly an algorithmic issue whereby finer resolution means more potential for noise/bias in the assumed surface reflectance relationship. What is the reason that going to a higher resolution makes things better for MISR? Is it a consequence of the way the land surface reflectance is modelled in the MISR standard algorithm, or does it suggest that 17.6 km was perhaps too coarse a resolution to use initially? The conclusion (page 13 lines 8-13) suggests the latter is the case, but I did not see direct evidence; it is definitely plausible, but I don't see why scene variability should lead to a persistent low AOD bias (as opposed to random noise) unless it's nonlinearity in the radiative transfer, or something in the way the algorithm partitions surface vs. atmospheric contributions to the satellite signal.

2. Throughout, the MISR/AERONET comparisons show AOD at 558 nm. AERONET provides spectral AOD and related quantities such as Ångström exponent, and in some cases retrievals of e.g. aerosol fine and coarse mode AOD. MISR retrieves AOD at 558 nm and a set of aerosol mixtures which fit the observations. These MISR aerosol mixtures have defined aerosol optical properties and so can be used to compute Ångström exponent or spectral AOD (and are often used to provide a categorical indication of aerosol 'type', which is one of MISR's selling points). The main focus of the aerosol data user community has been on midvisible AOD since this has been the main quan-
tity observed/retrieved by different techniques but it would be good to show similar types of plot for Ångström exponent and/or AOD at MISR's other wavelengths. If these also improve then it provides an indication that, for example, the set of aerosol mix-tures chosen by the retrieval is also improving, which is important for those interested in the 'aerosol type' applications of MISR data. This could be accomplished by adding analogues of Figure 4 for other wavelengths/ Ångström exponent.

3. The examples in this paper are drawn from AERONET DRAGON deployments. As the authors note, these are limited in geographical and temporal extent. There are a number of other areas where I think that the increase in spatial resolution might make a difference due to spatial heterogeneity on scales of a few km. For example, broken cloud fields (such as found in the Amazon) and near-source smoke or dust plumes (in many places of the world). It would be interesting to see a few examples of heterogeneous scenes like this (which don't necessarily have to be matched with AERONET sites) to see what the retrieval decides to do, both in terms of statistics of retrieved AOD, as well as whether a valid retrieval is obtained or not. This could have implications for aggregated statistics in level 3 products in some regions. If the authors would like some suggestions, I can provide some example MODIS Terra granules with interesting features (since MISR observes down the middle of MODIS Terra's swath).

4. A generalised danger in going to higher resolution is that artefacts can start appearing in a data set, due to contextual biases (e.g. related to surface cover) in the assumptions in the retrieval algorithm, leading to artificial structure in retrieved data fields which is taken to be real. This has been an issue for several other algorithms which operate at a higher resolution than the  $\sim$ 10 km scale common to most operational/heritage data products. In this case the DRAGON data suggest that, over these scenes at least, the bulk of the new finer-detail structure appearing in the MISR data is plausible. I would suggest adding a cautionary note to this effect to remind the reader of this possibility, perhaps around the end of the first paragraph in the conclusions where the 10 km/3km MODIS products are discussed, since this effect is not limited to
**the MODIS DT product.**

Specific comments:

It is a little tangential to the main point of the article, but the MODIS aerosol products' horizontal pixel sizes for the nominal 3 km and 10 km products are only valid near the centre of the MODIS swath. The broad swath and scan geometry mean that pixels get distorted in shape and size as the view zenith angle increases (often called the 'bow tie effect'), which makes them a lot larger than these nominal sizes and causes them to overlap, and in turn affects the characteristics of the level 2 data. See e.g. Wolfe et al (1998) and Sayer et al (2015b) for details. In contrast the MISR pixel size is, to my understanding, much less variable across-track.

Figures 2, 4: It would be good to add in plot titles or captions which data are being plotted here (i.e. California case for figure 2, all DRAGONs in Table 2 for figure 4.)

Figure 4: There are about a dozen points with AERONET AOD of 0.8 or higher, which are quite low-biased in the 17.6 km data set, but much closer to 1:1 in the 4.4 km data set. Are these from the same location or date, or more randomly distributed throughout the data set? This is relevant since, if they're from the same place or time, it could indicate that the higher resolution is particularly helpful for that specific circumstance, and it is interesting to know where you see a benefit vs. where it doesn't make much difference. From Table 2 I infer they may be from the Seoul deployment but it isn't clear whether they're the same date or from sites around Seoul itself (urban) or elsewhere in Korea. Same question for the outliers in more moderate-AOD cases (AERONET about 0.35, 0.4, 0.7; MISR about 0.15-0.2) which also jump more in-family when the retrieval is done at 4.4 km. This comment relates to my general comment 1 about figuring out why the higher resolution is helping.

Page 9, line 10: Can you expand a bit more on what 'complex terrain' means here? I guess it means variable-altitude scenes or similar, but a brief mention of what is tested for/how it is done (e.g. spectral/spatial tests, ancillary data base, etc) would be useful.
Figures 5, 6, 7: these are all on the same AOD colour scale, from 0-1.4. Figure 5 however is a much lower-AOD scene than the others, so it's hard to make out the patterns and values. Perhaps this could be redrawn on the same scale as Figure 1, i.e. 0-0.3? Also, for all these maps, it would be good if a different colour could be used for 'zero AOD' and 'no retrieval'; at the moment both are white. The colour bar font would benefit from being a little larger on all the maps (not legible on the pdf unless zoomed in).

Figure 6, 7 captions: delete 'the' in 'the Korea', or change to 'the Korean peninsula'.

Page 13, lines 14-19: I know that us data providers hate to hear the question, but if the authors are able to comment on whether there's a tentative schedule for the release of the new data set version, incorporating the higher resolution as well as the other updates mentioned in the referenced paragraph, that would be helpful. If it is up in the air then no need to include this.

References:

Sayer, A. M., Hsu, N. C., and Bettenhausen, C.: Implications of MODIS bow-tie distortion on aerosol optical depth retrievals, and techniques for mitigation, Atmos. Meas. Tech., 8, 5277-5288, doi:10.5194/amt-8-5277-2015, 2015b.

Wolfe, R. E., Roy, D., P., and Vermote, E.: MODIS Land Data Storage, Gridding, and Compositing Methodology: Level 2 Grid, IEEE Trans. Geosci. Remote Sens., 36, doi:10.1109/36.701082, 1998.

---

## Referee Comment (RC2) · Anonymous Referee #2 · 19 Aug 2016

The paper illustrates results of a prototype MISR algorithm at 4.4 km resolution, and demonstrates its improved performance with respect to the standard 17.6 km product with an assessment against relatively closely spaced DRAGON AERONET sites. As the authors point out, the availability of the DRAGON AERONET sites is a game-changer for enabling the assessment of the higher resolution product, and the performance is impressive and important to document. Overall, this is an interesting, well organized and easy-to-read paper. However, there are some areas where more details or clearer explanations would improve the manuscript. Some of these suggested additions are critically important, but since all suggested changes should be very easy for the authors to implement, they can be considered only minor revisions.

General comments:

The new MISR 4.4 aerosol product is mentioned for the first time in the same paragraph that describes the work of Kahn et al. 2010, Kalashnikova et al. 2013, etc. identifying specific performance issues with the V22 MISR algorithm. However, it is not stated whether these issues are addressed in the prototype 4.4 km algorithm or whether the prototype 4.4 km algorithm is different from V22 only in the resolution. In some parts of the manuscript, it seems clear that there are other changes besides just the resolution (for example, the bottom of page 9 where it is mentioned that the cost functions have been changed). However, in the discussion and conclusions, it states that the improvements did not require significant changes to the algorithm itself. It is very important to clarify and explain what algorithm differences there are between V22 and the prototype algorithm, and the mechanisms by which these changes lead to the observed improvements. This should be made clearer throughout the manuscript, in the introduction, methodology section, results, and discussion. The improvement is impressive regardless of whether it was solely due to the resolution change or not, but it's important for readers to understand how the algorithm changes produced the improvement.

Specific comments: Page 3, lines 7-11. The descriptions of the issues found by Kahn et al. (2010) should probably be expanded and clarified somewhat. What does "a small gap" mean? That description is evocative, but fairly ambiguous; I can think of several possible meanings. Similarly, what does "missing particles in the aerosol look up table" mean? Does this mean particle types? Does it mean that the particle types in the look up table did not adequately represent all observed aerosol types? Perhaps most importantly for the context of the current manuscript, was there any explanation (or speculation) for the systematic underestimate when AOD was greater than 0.4 (lines 10-11)?

Section 2: Figures 1 and 2 refer to a version of the 4.4 km prototype that was analyzed using the local mode data, whereas Figures 4-7 refer to a different version of the

prototype algorithm that uses different input data, at least. Please add some text early in section 2 mentioning that there are two different prototype algorithms, so it doesn't come as a surprise later in the section. Also, please make some distinction in the figure captions. Are there any other algorithm differences between these two versions besides what data is used for input? If so, make sure to describe them in the methods section.

Page 6, lines 11-30. There's a fairly ambiguous transition between the observation that the MODIS high resolution retrieval did not improve MODIS performance and the idea that the high resolution AERONET data is a requirement for adequate assessment of high resolution satellite products. The second paragraph makes a very good point about requiring a high resolution assessment data set. This paragraph starts neutrally "A further point", but do you mean to suggest that the assessment technique is part of the explanation for why the MISR high resolution product shows better performance and the MODIS high-res product didn't? After reading the conclusions, it seems that you are making this suggestion, so it should be made more explicit here where it is first brought up. Is the high resolution assessment the primary reason for the difference? If it is, then would a comparison of MISR 4.4 km with the "permanent" AERONET stations that MODIS used would also show little or no improvement? And would a comparison of MODIS 3 km product using the DRAGON sites be expected to show improvement? If this is not the primary explanation for the different results, do you have any explanation or theory what other factors are at play?

Section 2.3. Does this describe both the V22 algorithm and the prototype 4.4 km algorithm? Differences between them should be described here.

Page 9, line 26-27. "The fall-off evident in the V22 17.6 km resolution retrievals is greatly mitigated, if not eliminated entirely". Why? Please explain the mechanism by which going to higher resolution corrects a large bias at high AOD values. Or if there is more required than just the higher resolution, explain that. This is a critically important point of the paper and really needs to be explained well.

Page 9, line 30. "Relaxation of the thresholds on the chi-squared parameters to admit better spatial coverage". Relaxing the cost function seems like potentially a pretty significant change. Doesn't this mean that you are allowing the models to represent the aerosols a little less well than they do in V22? Would relaxing these thresholds also result in better spatial coverage in the V22 17.6 km resolution retrievals? This point seems like it needs more supporting material to understand its implications.

Page 10, line 12. When you say "the greatest benefit of the 4.4 km resolution MISR aerosol retrievals", it's not clear whether you mean the benefit of the higher resolution, or the benefit of the new prototype retrieval and all associated changes (of which the higher resolution is just one). Indeed, the better coverage is described as being due to the relaxation of the cost function, and not (or not primarily) due to the higher resolution, although later it is implied that it is due to the higher resolution because it can get in closer to exclusion zones.

Figures 2 and 4 are described as regressions both in the captions and the text, but there is no regression line shown, only a one-to-one line and prescribed error bars. It's important to show the regression lines if you describe this as a regression. Also consider including the slope in the statistics describing the regression (in the figure legend as well as the text). Are the RMSE values calculated with respect to the one-to-one line or the regression?

Technical comments:

Page 6, line 9. "Most significant improvements" (missing word)

Page 10, lines 27-30. These two sentences are both true but seem to give the opposite impression (high res has better coverage because of getting closer to exclusion zones; low res has better coverage because of fewer exclusion zones). So I suggest tweaking the wording and the transition between the two sentences. "In contrast" might make more sense than "for example".

The figures are too small to see the detail we are being directed to notice, without zooming in to 200% or even 400%. The AERONET data circles are not much bigger than a period in the figure caption and the color bar text is much, much smaller than the text in the caption. Please blow up the figures and remake the color bar text to make it easier on the reader.

---

## Referee Comment (RC3) · A. M. Sayer (Referee) · 22 Aug 2016

Dear authors and editor,

I agree with this second review's comments, with one caveat. The reviewer notes (correctly) that the scatter plots are referred to in the paper as 'regressions', but regression analysis results are not shown. The reviewer suggests either renaming them, or adding the regression slopes to the panels/discussion.

In my view renaming them is fine (and probably the best solution) but regression slopes should not be added. This is because linear least squares regression is an inappropriate tool to use for comparisons of this type, as these types of data sets violate the

statistical assumptions required for the technique to be valid. (Even though it is a commonly-used technique.)

---

## Author Response (AR1)

Response to A. M. Sayer (Referee)

We thank Dr. Sayer for his careful review of the manuscript and his useful comments. Below we provide specific responses to the comments. The reviewer's comments are in *italics*, and the responses are in normal text.

*1. As the authors note, the MODIS 3 km aerosol product was found to have poorer performance than the nominal 10 km product when compared to AERONET, which is the converse of the authors' experience with MISR, where the higher resolution improves things. My understanding is that with MODIS this is mostly an algorithmic issue whereby finer resolution means more potential for noise/bias in the assumed surface reflectance relationship. What is the reason that going to a higher resolution makes things better for MISR? Is it a consequence of the way the land surface reflectance is modelled in the MISR standard algorithm, or does it suggest that 17.6 km was perhaps too coarse a resolution to use initially? The conclusion (page 13 lines 8-13) suggests the latter is the case, but I did not see direct evidence; it is definitely plausible, but I don't see why scene variability should lead to a persistent low AOD bias (as opposed to random noise) unless it's nonlinearity in the radiative transfer, or something in the way the algorithm partitions surface vs. atmospheric contributions to the satellite signal.*

The reasons for the improvement of the MISR AOD retrievals when the spatial resolution is increased from 17.6 km to 4.4 km are complex. As the reviewer correctly notes, there are important fundamental differences between the MISR aerosol retrieval approach and the MODIS Dark Target (DT) or Deep Blue (DB) algorithms. The MODIS algorithms rely on assumed relationships in the surface spectral reflectances to account for the lower boundary condition. Overall, these relationships work well on a global basis, but are apparently adversely affected by the presence of noise, which increases as the resolution increases due to the reduction in the spatial averaging. The MISR retrieval approach, on the other hand, attempts to separate the angular contribution from the (assumed variable) surface and the overlying aerosols, which are assumed to be spatially homogeneous. To first order, when the aerosols are not spatially homogeneous – as in Figures 1, 6, and 7 – then this approach is likely to incorrectly assign this variability to the surface. This results in the surface contribution to the top of atmosphere radiances being overestimated, leading the algorithm to retrieve a lower AOD to compensate. This issue is explicitly described in the "Summary of Recommendations" in the assessment of the MISR V22 AODs by Kahn et al. (2010).

Going to higher resolution requires that the aerosols are spatially homogeneous on a much smaller spatial scale, so it is less likely that true aerosol variability is assigned to the surface, resulting in higher AODs. That said, even though the algorithms are identical, there are other consequences of changing the retrieval resolution that are more difficult to tease out. As the focus of this paper was on demonstrating the improvement in the MISR retrieved AODs relative to AERONET when the algorithm is run at a higher spatial resolution, rather than a complete description of the MISR retrieval algorithm, we felt it was out of scope to go into these details in the present work. It is our intention to further investigate these changes and report the results in a future publication.

Based on the suggestion of the both reviewers, we have added the following text to the manuscript to highlight the issue of aerosol variability and its effect on the retrieved AODs:

"Kahn et al. (2010) also identified a number of issues in the performance of the V22 MISR aerosol retrieval algorithm, including: lack of extremely low AODs in the MISR data compared to AERONET that causes an apparent "gap" in the comparison plots; the appearance of quantization noise; lack of particle types in the aerosol look up table to adequately represent all observed aerosol types; and a frequent underestimate of AOD relative to AERONET over land when the AOD was greater than about 0.4. The authors speculated that this underestimate was due to insufficiently absorbing particles being selected in cases where absorbing aerosols were present, or AOD variability at the 17.6 km spatial scale of the retrieval being incorrectly treated as surface variability reducing the contribution of aerosols to the top of atmosphere reflectances, resulting in a systematic underestimation of the AOD in these situations."

*2. Throughout, the MISR/AERONET comparisons show AOD at 558 nm. AERONET provides spectral AOD and related quantities such as Ångström exponent, and in some cases retrievals of e.g. aerosol fine and coarse mode AOD. MISR retrieves AOD at 558 nm and a set of aerosol mixtures which fit the observations. These MISR aerosol mixtures have defined aerosol optical properties and so can be used to compute Ångström exponent or spectral AOD (and are often used to provide a categorical indication of aerosol 'type', which is one of MISR's selling points). The main focus of the aerosol data user community has been on midvisible AOD since this has been the main quantity observed/retrieved by different techniques but it would be good to show similar types of plot for Ångström exponent and/or AOD at MISR's other wavelengths. If these also improve then it provides an indication that, for example, the set of aerosol mixtures chosen by the retrieval is also improving, which is important for those interested in the 'aerosol type' applications of MISR data. This could be accomplished by adding analogues of Figure 4 for other wavelengths/ Ångström exponent.*

As Dr. Sayer is no doubt aware, and as described in more detail in Kahn and Gaitley (2015), particle property validation using AERONET requires specific aerosol loading and viewing conditions that are infrequently realized, particularly for a small sample size such as the AERONET-DRAGON cases discussed in this work. Ångström exponent comparisons, by their nature, are fundamentally qualitative because they relate spectral slopes that can vary significantly with even small changes in retrieved spectral AOD. The assessments presented in this work are specifically for midvisible AOD so, to avoid confusion, we have changed the title of the paper to "Development and Assessment of a Higher Spatial Resolution (4.4 km) MISR Aerosol Optical Depth Product Using AERONET-DRAGON Data." We have also made changes throughout the manuscript to highlight that this work is a comparison of AOD only. As part of the algorithm development for the new (V23) MISR aerosol product, we plan to assess the particle property information and present these results in a future publication.

*3. The examples in this paper are drawn from AERONET DRAGON deployments. As the authors note, these are limited in geographical and temporal extent. There are a number of other areas where I think that the increase in spatial resolution might make a difference due to spatial heterogeneity on scales of a few km. For example, broken cloud fields (such as found in the Amazon) and near-source smoke or dust plumes (in many places of the world). It would be interesting to see a few examples of heterogeneous scenes like this (which don't necessarily have to be matched with AERONET sites) to see what the retrieval decides to do, both in terms of statistics of retrieved AOD, as well as whether a valid retrieval is obtained or not. This could have implications for aggregated statistics in level 3 products in some regions. If the authors would like some suggestions, I can provide some example MODIS Terra granules with interesting features (since MISR observes down the middle of MODIS Terra's swath).*

Dr. Sayer makes some excellent suggestions for examining the performance of the MISR aerosol retrieval in heterogeneous scenes. Broken cloud fields and near-source plumes are of particular scientific interest. The work presented here was done to be included as part of an ACP/AMT special issue on "Meso-scale aerosol processes, comparison and validation studies from DRAGON networks." This is the reason for the specific focus on the AERONET-DRAGON results. That said, we would be very interest to get a set of cases from Dr. Sayer that could be examined as part of a more comprehensive retrieval validation effort.

*4. A generalised danger in going to higher resolution is that artefacts can start appearing in a data set, due to contextual biases (e.g. related to surface cover) in the assumptions in the retrieval algorithm, leading to artificial structure in retrieved data fields which is taken to be real. This has been an issue for several other algorithms which operate at a higher resolution than the ~10 km scale common to most operational/heritage data products. In this case the DRAGON data suggest that, over these scenes at least, the bulk of the new finer-detail structure appearing in the MISR data is plausible. I would suggest adding a cautionary note to this effect to remind the reader of this possibility, perhaps around the end of the first paragraph in the conclusions where the 10 km/3km MODIS products are discussed, since this effect is not limited to the MODIS DT product.*

This is an important point. The resolution of satellite retrievals is often dictated by the need to mitigate the effects of noise in the instrument observations. Retrievals are then built that contain assumptions about the behavior of the atmosphere and/or surface that seem to be appropriate for these spatial scales. When the scale of the retrieval is changed, these assumptions may no longer be appropriate, leading to unexpected retrieval results. To address this, the following has been add to the text immediately following the first paragraph in the "Discussion and conclusions" section:

"Simply providing results at a higher spatial resolution does not guarantee an improvement in the performance of a satellite retrieval algorithm. From a remote sensing standpoint, observations are typically averaged over some spatial scale in an attempt to reduce the impact of random noise in the observations themselves. Changes to the resolution can introduce unexpected biases due to changes in the assumptions (e.g., spatial homogeneity, spectral relationships) developed and implemented for coarser resolution retrievals."

*It is a little tangential to the main point of the article, but the MODIS aerosol products' horizontal pixel sizes for the nominal 3 km and 10 km products are only valid near the centre of the MODIS swath. The broad swath and scan geometry mean that pixels get distorted in shape and size as the view zenith angle increases (often called the 'bow tie effect'), which makes them a lot larger than these nominal sizes and causes them to overlap, and in turn affects the characteristics of the level 2 data. See e.g. Wolfe et al (1998) and Sayer et al (2015b) for details. In contrast the MISR pixel size is, to my understanding, much less variable across-track.*

The difference in the MODIS swath (2,330 km) compared to the MISR swath (380 km for the nadir camera) shows that the change in the pixel size in the cross-track direction is a much larger issue for MODIS than it is for MISR.  MISR is also a pushbroom sensor, compared to the whiskbroom MODIS sensor, so, again, the effects of the cross-track viewing geometry are smaller for MISR than they are for MODIS.

*Figures 2, 4: It would be good to add in plot titles or captions which data are being plotted here (i.e. California case for figure 2, all DRAGONs in Table 2 for figure 4.)*

The suggested changes were made.

*Figure 4: There are about a dozen points with AERONET AOD of 0.8 or higher, which are quite low-biased in the 17.6 km data set, but much closer to 1:1 in the 4.4 km data set. Are these from the same location or date, or more randomly distributed throughout the data set? This is relevant since, if they're from the same place or time, it could indicate that the higher resolution is particularly helpful for that specific circumstance, and it is interesting to know where you see a benefit vs. where it doesn't make much difference. From Table 2 I infer they may be from the Seoul deployment but it isn't clear whether they're the same date or from sites around Seoul itself (urban) or elsewhere in Korea. Same question for the outliers in more moderate-AOD cases (AERONET about 0.35, 0.4, 0.7; MISR about 0.15-0.2) which also jump more in-family when the retrieval is done at 4.4 km. This comment relates to my general comment 1 about figuring out why the higher resolution is helping.*

In spite of the range of AERONET-DRAGON deployments, the number of mostly cloud-free coincidences with MISR is fairly low.  Table 2 illustrates the problem.  This is the complete set of MISR/AERONET-DRAGON matchups that were identified.  The AODs tended to be stratified, with the highest AOD cases being from Asia-Seoul.  Figure 6 shows that the highest AODs were observed around Seoul on 9 May 2012.  High AODs were also observed elsewhere in Korea on both 9 May and 25 May 2012.  Page 11, lines 23-27 in the manuscript describe these cases.

What is particularly striking in Figure 4 is the overall elimination of low outliers.  This seems to support the assertion that increasing the spatial resolution has the effect of reducing the low AOD bias apparent in the V22 algorithm results.

*Page 9, line 10: Can you expand a bit more on what 'complex terrain' means here? I guess it means variable-altitude scenes or similar, but a brief mention of what is tested for/how it is done (e.g. spectral/spatial tests, ancillary data base, etc) would be useful.*

The text was modified as follows: "No retrievals are performed over complex terrain (i.e., where the standard deviation of the regional surface elevation exceeds 500 m based on the MISR digital elevation model)."

*Figures 5, 6, 7: these are all on the same AOD colour scale, from 0-1.4. Figure 5 however is a much lower-AOD scene than the others, so it's hard to make out the patterns and values. Perhaps this could be redrawn on the same scale as Figure 1, i.e. 0-0.3? Also, for all these maps, it would be good if a different colour could be used for 'zero AOD' and 'no retrieval'; at the moment both are white. The colour bar font would benefit from being a little larger on all the maps (not legible on the pdf unless zoomed in).*

The color scale was deliberately kept the same for all three figures to facilitate intercomparison of the cases. The spatial variability of the AOD in Figure 5 is really quite limited, so changing the scale does not reveal very much additional detail. For MISR (unlike MODIS) a zero AOD is actually a missing retrieval (i.e., if the algorithm retrieves an AOD of 0.0, then this is considered a "failed" retrieval). Only pixels in color indicate successful retrievals. The overall size of the color bar has been increased in the revised manuscript, hopefully improving legibility.

*Figure 6, 7 captions: delete 'the' in 'the Korea', or change to 'the Korean peninsula'.*

This was a typo, thank you for catching it.

*Page 13, lines 14-19: I know that us data providers hate to hear the question, but if the authors are able to comment on whether there's a tentative schedule for the release of the new data set version, incorporating the higher resolution as well as the other updates mentioned in the referenced paragraph, that would be helpful. If it is up in the air then no need to include this.*

The current plan is to deliver the updated algorithm to NASA Langley for processing in Spring 2017. The text has been modified as follows: "The MISR aerosol algorithm team is working toward the release of an updated version of the aerosol retrieval in Spring 2017 that will have results reported globally at 4.4 km resolution.

Response to Anonymous Referee #2

We thank the referee for this careful review of the manuscript and the suggestions to improve the clarity of the work. Below we provide specific responses to the comments. The reviewer's comments are in *italics*, and the responses are in normal text.

*The new MISR 4.4 aerosol product is mentioned for the first time in the same paragraph that describes the work of Kahn et al. 2010, Kalashnikova et al. 2013, etc. identifying specific performance issues with the V22 MISR algorithm. However, it is not stated whether these issues are addressed in the prototype 4.4 km algorithm or whether the prototype 4.4 km algorithm is different from V22 only in the resolution. In some parts of the manuscript, it seems clear that there are other changes besides just the resolution (for example, the bottom of page 9 where it is mentioned that the cost functions have been changed). However, in the discussion and conclusions, it states that the improvements did not require significant changes to the algorithm itself. It is very important to clarify and explain what algorithm differences there are between V22 and the prototype algorithm, and the mechanisms by which these changes lead to the observed improvements. This should be made clearer throughout the manuscript, in the introduction, methodology section, results, and discussion. The improvement is impressive regardless of whether it was solely due to the resolution change or not, but it's important for readers to understand how the algorithm changes produced the improvement.*

There are no significant changes to the algorithm used for the 4.4 km retrievals compared to the 17.6 km retrievals. The relevant changes have to do with the input data, both in terms of resolution when discussing the "local mode" data, and the area of interest (4.4 km vs. 17.6 km). The relaxation of the $\chi^2$ threshold described on page 9 of the manuscript refers to a decision regarding whether or not a specific retrieval was considered "successful." This primarily impacts the coverage obtained by the 4.4 km algorithm and the adjustment was required because the threshold was designed to provide adequate coverage for the original 17.6 km product. The manuscript has been modified to make the equivalence between the 4.4 km and 17.6 km algorithm more apparent throughout per the reviewer's suggestion.

*Specific comments: Page 3, lines 7-11. The descriptions of the issues found by Kahn et al. (2010) should probably be expanded and clarified somewhat. What does "a small gap" mean? That description is evocative, but fairly ambiguous; I can think of several possible meanings. Similarly, what does "missing particles in the aerosol look up table" mean? Does this mean particle types? Does it mean that the particle types in the look up table did not adequately represent all observed aerosol types? Perhaps most importantly for the context of the current manuscript, was there any explanation (or speculation) for the systematic underestimate when AOD was greater than 0.4 (lines 10-11)?*

The list on Page 3, lines 7-11 was meant to provide a summary of the issues identified in the V22 MISR aerosol product, with the idea that an interested reader would be able to find more information in the papers themselves. However, in the interest of making our paper more self-contained the section has been modified as follows:

"Kahn et al. (2010) also identified a number of issues in the performance of the V22 MISR aerosol retrieval algorithm, including: lack of extremely low AODs in the MISR data compared to AERONET that causes an apparent "gap" in the comparison plots; the appearance of quantization noise; lack of particle types in the aerosol look up table to adequately represent all observed aerosol types; and a frequent underestimate of AOD relative to AERONET over land when the AOD was greater than about 0.4. The authors speculated that this underestimate was due to insufficiently absorbing particles being selected in cases where absorbing aerosols were present, or AOD variability at the 17.6 km spatial scale of the retrieval being incorrectly treated as surface variability reducing the contribution of aerosols to the top of atmosphere reflectances, resulting in a systematic underestimation of the AOD in these situations."

*Section 2: Figures 1 and 2 refer to a version of the 4.4 km prototype that was analyzed using the local mode data, whereas Figures 4-7 refer to a different version of the prototype algorithm that uses different input data, at least. Please add some text early in section 2 mentioning that there are two different prototype algorithms, so it doesn't come as a surprise later in the section. Also, please make some distinction in the figure captions. Are there any other algorithm differences between these two versions besides what data is used for input? If so, make sure to describe them in the methods section.*

Again, the key point is that the *algorithms* are identical. The input data are different, however. The figure captions have been updated per the suggestions of both reviewers to make their content clearer.

*Page 6, lines 11-30. There's a fairly ambiguous transition between the observation that the MODIS high resolution retrieval did not improve MODIS performance and the idea that the high resolution AERONET data is a requirement for adequate assessment of high resolution satellite products. The second paragraph makes a very good point about requiring a high resolution assessment data set. This paragraph starts neutrally "A further point", but do you mean to suggest that the assessment technique is part of the explanation for why the MISR high resolution product shows better performance and the MODIS high-res product didn't? After reading the conclusions, it seems that you are making this suggestion, so it should be made more explicit here where it is first brought up. Is the high resolution assessment the primary reason for the difference? If it is, then would a comparison of MISR 4.4 km with the "permanent" AERONET stations that MODIS used would also show little or no improvement? And would a comparison of MODIS 3 km product using the DRAGON sites be expected to show improvement? If this is not the primary explanation for the different results, do you have any explanation or theory what other factors are at play?*

The existence of the MODIS 3 km data and the conclusions drawn by Remer et al. (2013) create unexpected difficulties for this work. As the reviewer correctly points out, the Remer et al. (2013) analysis is for a globally distributed set of AERONET sites, which does not include AERONET-DRAGON deployments. Munchak et al. (2013) do compare the 3 km MODIS Dark Target (MODIS-DT) results with AERONET-DRAGON

in the Washington, D.C./Baltimore area. They identified other issues with the MODIS-DT algorithm having to do with urban areas violating the Dark Target algorithm assumptions. A comparison of the MISR 4.4 km AOD retrievals with a larger suite of AERONET sites is ongoing. The primary factor in the improvement of the 4.4 km MISR product relative to the 17.6 km product likely has to do with the assumption of aerosol spatial variability on these different scales. As mentioned above, the algorithm attempts to separate the surface (assumed to be heterogeneous) from the aerosol (assumed to be homogeneous). It seems that 4.4 km is a more appropriate spatial scale for assumed aerosol homogeneity than 17.6 km, at least for the MISR retrieval.

*Section 2.3. Does this describe both the V22 algorithm and the prototype 4.4 km algorithm? Differences between them should be described here.*

As mentioned above, the algorithms are the same, so this section describes both the 17.6 km and 4.4 km retrieval algorithms. The text has been modified elsewhere as suggested to highlight their equivalence.

*Page 9, line 26-27. "The fall-off evident in the V22 17.6 km resolution retrievals is greatly mitigated, if not eliminated entirely". Why? Please explain the mechanism by which going to higher resolution corrects a large bias at high AOD values. Or if there is more required than just the higher resolution, explain that. This is a critically important point of the paper and really needs to be explained well.*

As noted in the response to the other reviewer, the reasons for the improvement in the MISR retrievals at 4.4 km compared to 17.6 km are complex. Going to higher resolution requires that the aerosols are spatially homogeneous on a much smaller spatial scale, so it is less likely that true aerosol variability is assigned to the surface, resulting in higher AODs. That said, even though the algorithms are identical, there are other consequences of changing the retrieval resolution that are more difficult to tease out. As the focus of this paper was on demonstrating the improvement in the MISR retrieved AODs relative to AERONET when the algorithm is run at a higher spatial resolution, rather than a complete description of the MISR retrieval algorithm, we felt it was out of scope to go into these details in the present work. It is our intention to further investigate these changes and report the results in a future publication.

*Page 9, line 30. "Relaxation of the thresholds on the chi-squared parameters to admit better spatial coverage". Relaxing the cost function seems like potentially a pretty significant change. Doesn't this mean that you are allowing the models to represent the aerosols a little less well than they do in V22? Would relaxing these thresholds also result in better spatial coverage in the V22 17.6 km resolution retrievals? This point seems like it needs more supporting material to understand its implications.*

As previously mentioned, the relaxation of the $\chi^2$ threshold is necessary to maintain the spatial coverage of the 4.4 km product relative to the 17.6 km product for which the threshold was initially developed. While it is true that this effectively allows the 4.4 km retrieval to be successful for an AOD/aerosol model combination that agrees with the observations less well than in the case of the 17.6 km retrieval, the choice of the threshold was made somewhat arbitrarily (i.e., "tuned") to provide good coverage at 17.6 km resolution.  Making a similar change to the 17.6 km retrievals has comparatively little effect on the coverage.

*Page 10, line 12. When you say "the greatest benefit of the 4.4 km resolution MISR aerosol retrievals", it's not clear whether you mean the benefit of the higher resolution, or the benefit of the new prototype retrieval and all associated changes (of which the higher resolution is just one). Indeed, the better coverage is described as being due to the relaxation of the cost function, and not (or not primarily) due to the higher resolution, although later it is implied that it is due to the higher resolution because it can get in closer to exclusion zones.*

The sentence refers to the 4.4 km resolution product including the associated changes in the $\chi^2$ threshold.  We are comparing the results of the 17.6 km algorithm (as implement in the operational V22 MISR aerosol product) with the 4.4 km algorithm results in an overall sense.

*Figures 2 and 4 are described as regressions both in the captions and the text, but there is no regression line shown, only a one-to-one line and prescribed error bars. It's important to show the regression lines if you describe this as a regression. Also consider including the slope in the statistics describing the regression (in the figure legend as well as the text). Are the RMSE values calculated with respect to the one- to-one line or the regression?*

The reviewer is correct that the term "regression" was used inappropriately for these intercomparisons.  The text has been changed to "scatterplots" or "intercomparisons" as appropriate.  There are no linear regressions performed for reasons clearly elucidated by Dr. Sayer in his comment on this issue.  The RMSE values are calculated with respect to the paired AERONET values.

*Page 6, line 9. "Most significant improvements" (missing word)*

Changed.

*Page 10, lines 27-30. These two sentences are both true but seem to give the opposite impression (high res has better coverage because of getting closer to exclusion zones; low res has better coverage because of fewer exclusion zones). So I suggest tweaking the wording and the transition between the two sentences. "In contrast" might make more sense than "for example".*

This is a good suggestion: "for example" has been changed to "in contrast".

*The figures are too small to see the detail we are being directed to notice, without zooming in to 200% or even 400%. The AERONET data circles are not much bigger than a period in the figure caption and the color bar text is much, much smaller than the text*

*in the caption. Please blow up the figures and remake the color bar text to make it easier on the reader.*

We have made the colorbars larger in the revised manuscript to hopefully improve the legibility.

Manuscript Changes

**1. Introduction**

*The following text was added:*

Kahn et al. (2010) also identified a number of issues in the performance of the V22 MISR aerosol retrieval algorithm, including: lack of extremely low AODs in the MISR data compared to AERONET that causes an apparent "gap" in the comparison plots; the appearance of quantization noise; lack of particle types in the aerosol look up table to adequately represent all naturally occurring aerosol types; and a frequent underestimate of AOD relative to AERONET over land when the AOD is greater than about 0.4. The authors speculated that this underestimate was due to insufficiently absorbing particles being selected in cases where absorbing aerosols were present, or AOD variability at the 17.6 km spatial scale of the retrieval being incorrectly treated as surface variability reducing the contribution of aerosols to the top-of-atmosphere reflectances, resulting in a systematic underestimation of the AOD in these situations.

As we will discuss in this paper, we found that a MISR 4.4 aerosol retrieval using the same algorithm as the operational (V22) 17.6 km product is better able to resolve spatial gradients in AOD as shown in a number of comparisons from different DRAGON deployments that encompass a wide range of aerosol loadings.

**2.2 AERONET-DRAGON Deployments**

*The following text was added:*

Additionally, certain AOD ranges occur preferentially for different DRAGON deployments, with the highest AODs occurring in South Korea.

**2.3 MISR aerosol retrievals over land**

*The following text was added:*

> 6. No retrievals are performed over complex terrain (i.e., where the standard deviation of the regional surface elevation exceeds 500 m based on the MISR digital elevation model).

Note that for the comparisons shown in the next section, the aerosol retrieval algorithm was not modified except to provide results at 4.4 km, as opposed to the 17.6 km resolution of the operational retrieval, and the absolute threshold on the $\chi^2$ parameter was relaxed to provide a better match to the coverage of the 17.6 km product. This was required because the value of this threshold was tuned for the 17.6 km product and the coverage of the 4.4 km retrievals was significantly worse in some cases. If anything, adjusting this threshold for the 4.4 km retrievals will allow aerosol models with poorer agreement with the MISR observations to be considered successful.

**4. Discussion and conclusions**

*The following text was added:*

Simply providing results at a higher spatial resolution does not guarantee an improvement in the performance of a satellite retrieval algorithm, however. From a remote sensing standpoint, observations are typically averaged over some spatial scale in an attempt to reduce the impact of random noise in the observations themselves. Changes to the resolution can introduce unexpected biases due to changes in the assumptions (e.g., spatial homogeneity, spectral relationships) developed and implemented for coarser resolution retrievals.

**Development and Assessment of a Higher Spatial Resolution (4.4 km) MISR Aerosol Optical Depth Product Using AERONET-DRAGON Data**

**Michael J. Garay[1], Olga V. Kalashnikova[1], and Michael A. Bull[1]**

[revised manuscript text omitted]

aerosol retrieval algorithm, including: lack of extremely low AODs in the MISR data compared to AERONET that causes an apparent "gap" in the comparison plots; the appearance of quantization noise; lack of particle types in the aerosol look up table to adequately represent all naturally occurring aerosol types; and a frequent underestimate of

AOD relative to AERONET over land when the AOD is greater than about 0.4. The authors speculated that this underestimate was due to insufficiently absorbing particles being selected in cases where absorbing aerosols were present, or AOD variability at the 17.6 km spatial scale of the retrieval being incorrectly treated as surface variability reducing the contribution of aerosols to the top-of-atmosphere reflectances, resulting in a systematic underestimation of the AOD in these situations. Subsequently, Kalashnikova et al. (2013), Witek et al. (2013), and Shi et al. (2014) identified issues with the cloud screening applied in the V22 algorithm, especially with regard to thin cirrus, and suggested possible solutions; and Limbacher and

Kahn (2015) diagnosed the effects of stray light in the MISR cameras, noted earlier by

Bruegge et al. (2002), that could have significant impact on retrieved AODs in scenes with high contrast. These efforts by members of the MISR science team and others have been directed at improving the quality of the MISR aerosol product with the view of delivering a new version of the operational MISR aerosol retrieval algorithm in the near future. At the same time, a number of studies have highlighted the need for aerosol products at higher spatial resolutions than currently available operationally from MISR and MODIS, gridded at

17.6 km and 10.0 km, respectively. In response to this, the MODIS team released a global 3

km resolution DT aerosol product as part of its Collection 6 delivery (Remer et al., 2013). In this work, we describe the effort to develop a higher resolution, 4.4 km Level 2 MISR aerosol product based on initial tests that showed significant AOD retrieval improvement relative to

AERONET sites deployed in relatively large numbers locally in Distributed Regional Aerosol

Gridded Observations Network (DRAGON) campaigns in regions around the globe (e.g., Eck et al., 2014; Seo et al., 2015; Sano et al., 2016). In the DRAGON networks, instruments are located much closer to one another, with a typical grid spacing around 10 km (e.g., Munchak et al., 2013). As we will discuss in this paper, we found that a MISR 4.4 aerosol retrieval using the same algorithm as the operational (V22) 17.6 km product is better able to resolve

Michael J Garay 2/13/2017 5:24 PM

Michael J Garay 2/13/2017 5:24 PM

Michael J Garay 2/13/2017 5:25 PM

Michael J Garay 2/13/2017 6:42 PM

Michael J Garay 2/13/2017 6:47 PM

Michael J Garay 2/13/2017 6:47 PM

Michael J Garay 2/13/2017 6:47 PM

[revised manuscript text omitted]

6. No retrievals are performed over complex terrain (i.e., where the standard deviation of the regional surface elevation exceeds 500 m based on the MISR digital elevation model).

The choice of acceptable 1.1 km subregions within the retrieval region is done through the application of a number of tests including cloud masking. Note that for the comparisons shown in the next section, the aerosol retrieval algorithm was not modified except to provide results at 4.4 km, as opposed to the 17.6 km resolution of the operational retrieval, and the absolute threshold on the $\chi^2$ parameter was relaxed to provide a better match to the coverage of the 17.6 km product. This was required because the value of this threshold was tuned for the 17.6 km product and the coverage of the 4.4 km retrievals was significantly worse in some cases. If anything, adjusting this threshold for the 4.4 km retrievals will allow aerosol models with poorer agreement with the MISR observations to be considered successful.

**3   Results**

**3.1   AOD comparison plots**

Figure 4a shows the comparison of the V22 17.6 km MISR green-band AODs against the AERONET-DRAGON AODs interpolated to the MISR wavelength (558 nm) for all the cases listed in Table 2. The range of AODs in this figure is much greater than the AOD range in Fig. 2. Like the comparisons shown in Kahn et al. (2010) the underestimation of the retrieved AODs relative to AERONET for AODs greater than about 0.4 is apparent in this figure. By way of comparison, Fig. 4b shows the results for a prototype 4.4 km MISR aerosol retrieval (internally designated V22b24-34+1) that takes the 1.1 km spatial resolution global mode data as input. Tests showed that the AOD retrievals from this algorithm were not significantly different from the AODs retrieved using the 275 m local mode data as input.

[revised manuscript text omitted]

Michael J Garay 2/14/2017 12:24 PM

Michael J Garay 2/14/2017 12:25 PM

Key to the development of this new algorithm are assessments against a range of cases represented by those used in this paper.

**Acknowledgements**

This work was performed at the Jet Propulsion Laboratory, California Institute of Technology under a contract with the National Aeronautics and Space Administration.  The MISR data used in this work were obtained from the NASA Langley Research Center Atmospheric

Science Data Center.  We thank the many PI investigators, and particularly the hard work of

Brent Holben and his team for establishing and maintaining the AERONET and AERONET-

DRAGON sites used in this investigation.  We are also grateful to Dr. Andrew Sayer and an anonymous reviewer for their thoughtful comments that have helped improve this paper.

[revised manuscript text omitted]

---

## Author Response (AR2)

Manuscript Changes

**Abstract**

*The following text was added:*

In comparisons with AERONET-DRAGON AODs the 4.4 km resolution retrievals show
improved correlation ($r = 0.9595$), smaller root mean squared error (0.0768), reduced bias (-
0.0208), and a larger fraction within the expected error envelope (80.92%) relative to the
Version 22 MISR retrievals.

**4. Discussion and conclusions**

*The following text was added:*

The reasons for the improvement of the MISR AOD retrievals when the spatial resolution is
increased from 17.6 km to 4.4 km are complex.  The MODIS algorithms rely on assumed
relationships in the surface spectral reflectances to account for the lower boundary condition
(Levy et al., 2013).   Overall, these relationships work well on a global basis, but are
apparently adversely affected by the presence of noise, which increases as the resolution
increases due to reduction in the spatial averaging.  The MISR retrieval approach, on the other
hand, attempts to separate the angular contribution from the (assumed variable) surface and
the overlying aerosols, which are assumed to be spatially homogeneous (Martonchik et al.,
2009).  To first order, when the aerosols are not spatially homogeneous then this approach is
likely to incorrectly assign this variability to the surface. Kahn et al. (2010) hypothesize that
this results in the surface contribution to the top of atmosphere radiances being overestimated,
leading the algorithm to retrieve a lower AOD to compensate.

From a remote sensing standpoint, observations are typically averaged over some spatial scale
in an attempt to reduce the impact of random noise in the observations themselves, as in the
case of MODIS retrievals.

In comparisons with AERONET-DRAGON AODs for a variety of globally distributed
deployments the 4.4 km resolution retrievals show improved correlation ($r = 0.9595$), smaller
root mean squared error (0.0768), reduced bias (-0.0208), and a larger fraction within the
expected error envelope (80.92%). The results for the V22 algorithm are $r = 0.8772$, RMSE =
0.1683, bias = -0.0887, and 59.09% in the expected error envelope, as shown in Fig. 4.

**Figures**

Figures 1, 5, 6, and 7 were revised per the reviewers' suggestions to make the contents and
captions more legible.

**Development and Assessment of a Higher Spatial Resolution (4.4 km) MISR Aerosol Optical Depth Product Using AERONET-DRAGON Data**

**Michael J. Garay[1], Olga V. Kalashnikova[1], and Michael A. Bull[1]**

[revised manuscript text omitted]

6. No retrievals are performed over complex terrain (i.e., where the standard deviation of the regional surface elevation exceeds 500 m based on the MISR digital elevation model).

The choice of acceptable 1.1 km subregions within the retrieval region is done through the application of a number of tests including cloud masking. Note that for the comparisons shown in the next section, the aerosol retrieval algorithm was not modified except to provide results at 4.4 km, as opposed to the 17.6 km resolution of the operational retrieval, and the absolute threshold on the $\chi^2$ parameter was relaxed to provide a better match to the coverage of the 17.6 km product. This was required because the value of this threshold was tuned for the 17.6 km product and the coverage of the 4.4 km retrievals was significantly worse in some cases. If anything, adjusting this threshold for the 4.4 km retrievals will allow aerosol models with poorer agreement with the MISR observations to be considered successful.

**3 Results**

**3.1 AOD comparison plots**

Figure 4a shows the comparison of the V22 17.6 km MISR green-band AODs against the AERONET-DRAGON AODs interpolated to the MISR wavelength (558 nm) for all the cases listed in Table 2. The range of AODs in this figure is much greater than the AOD range in Fig. 2. Like the comparisons shown in Kahn et al. (2010) the underestimation of the retrieved AODs relative to AERONET for AODs greater than about 0.4 is apparent in this figure. By way of comparison, Fig. 4b shows the results for a prototype 4.4 km MISR aerosol retrieval (internally designated V22b24-34+1) that takes the 1.1 km spatial resolution global mode data as input. Tests showed that the AOD retrievals from this algorithm were not significantly different from the AODs retrieved using the 275 m local mode data as input.

[revised manuscript text omitted]

The biggest surprise in moving the aerosol retrieval to a higher spatial resolution was the improvement in the retrieved AOD relative to AERONET – an improvement that did not require changes to the algorithm itself. This was surprising for two reasons. First, the more or less accepted line of thought was that aerosols are generally spatially homogeneous at scales of 10's to 100's of kilometers, and temporally stationary, in a statistical sense, at time scales of hours to days (e.g., Anderson et al., 2003).  Secondly, the MODIS team did not find significant improvement in the performance of their algorithm when they increased the resolution from 10 km to 3 km (Remer et al., 2013).  In fact, this change in resolution highlighted some underlying issues in the assumptions going into the DT retrieval (Munchak et al., 2013).  The reasons for the improvement of the MISR AOD retrievals when the spatial resolution is increased from 17.6 km to 4.4 km are complex.  The MODIS algorithms rely on assumed relationships in the surface spectral reflectances to account for the lower boundary condition (Levy et al., 2013).  Overall, these relationships work well on a global basis, but are apparently adversely affected by the presence of noise, which increases as the resolution increases due to reduction in the spatial averaging.  The MISR retrieval approach, on the other hand, attempts to separate the angular contribution from the (assumed variable) surface and the overlying aerosols, which are assumed to be spatially homogeneous (Martonchik et al., 2009).  To first order, when the aerosols are not spatially homogeneous then this approach is likely to incorrectly assign this variability to the surface. Kahn et al. (2010) hypothesize that this results in the surface contribution to the top of atmosphere radiances being overestimated, leading the algorithm to retrieve a lower AOD to compensate.

Simply providing results at a higher spatial resolution does not guarantee an improvement in the performance of a satellite retrieval algorithm, however.  From a remote sensing standpoint, observations are typically averaged over some spatial scale in an attempt to reduce the impact of random noise in the observations themselves, as in the case of MODIS retrievals.  Changes to the resolution can introduce unexpected biases due to changes in the assumptions (e.g., spatial homogeneity, spectral relationships) developed and implemented for coarser resolution retrievals.  Importantly, it would have been difficult to assess the performance of a high-resolution algorithm without appropriate high-resolution observations to evaluate against. A single AERONET site basically returns a "point" in space and time relative to retrievals from a satellite instrument.  This has led to the adoption of averaging approaches that require large amounts of paired satellite-AERONET data matched within relative broad spatial and temporal windows (e.g., Ichoku et al., 2003; Kahn et al., 2010; Petrenko et al., 2012).  The deployment of AERONET-DRAGON sites beginning in 2011 has been a game-changer in terms of the ability to truly consider aerosol spatial variability and the DRAGON deployments at sites around the globe facilitated the analysis presented here.

The performance of the operational V22 17.6 km MISR aerosol retrieval relative to the performance of a prototype 4.4 km retrieval was assessed in comparisons with multiple AERONET-DRAGON deployments over a broad range of AODs. It was found that, overall, the 4.4 km AOD retrieval performed significantly better than the 17.6 km retrieval. In comparisons with AERONET-DRAGON AODs for a variety of globally distributed deployments the 4.4 km resolution retrievals show improved correlation ($r = 0.9595$), smaller root mean squared error (0.0768), reduced bias (-0.0208), and a larger fraction within the expected error envelope (80.92%). The results for the V22 algorithm are $r = 0.8772$, RMSE = 0.1683, bias = -0.0887, and 59.09% in the expected error envelope, as shown in Fig. 4. Part of the reason for this improvement is the ability of the higher-resolution retrieval to capture the true spatial variability of the aerosols, which is also captured by the DRAGON networks. Again, a single AERONET site cannot directly represent the spatial variability of aerosols, although this is aliased into the temporal dependence of the AOD observed by the instrument. Averaging the AERONET data over a time window and the satellite data over a spatial window, as is traditionally done in global comparisons, has the effect of minimizing the contributions of true aerosol spatial variability. Another reason for the improvement of the MISR retrieval algorithm when applied at 4.4 km is that the assumptions underlying the aerosol retrieval, particularly over land, are better met at this higher spatial resolution. Ironically, among the most critical of these assumptions is that aerosols are spatially homogeneous on the scale of the retrieval. In other words, aerosol variability itself is likely one of the issues with the 17.6 km retrieval.

The MISR aerosol algorithm team is working toward the release of an updated version of the aerosol retrieval in Spring 2017 that will have results reported globally at 4.4 km resolution. In addition to this change, other changes are being tested and implemented with regard to cloud screening, per-retrieval uncertainty reporting, and microphysical property retrievals. Key to the development of this new algorithm are assessments against a range of cases represented by those used in this paper.

**Acknowledgements**

This work was performed at the Jet Propulsion Laboratory, California Institute of Technology under a contract with the National Aeronautics and Space Administration. The MISR data used in this work were obtained from the NASA Langley Research Center Atmospheric

Michael J Garay 3/10/2017 7:19 PM

Science Data Center.  We thank the many PI investigators, and particularly the hard work of

Brent Holben and his team for establishing and maintaining the AERONET and AERONET-

DRAGON sites used in this investigation.  We are also grateful to Dr. Andrew Sayer and an anonymous reviewer for their thoughtful comments that have helped improve this paper.

[revised manuscript text omitted]

DRAGON interpolated to the MISR wavelength for 20 January 2013. (b) Comparison of the

4.4 km resolution AODs against AERONET-DRAGON for the same date.

[Figure]

Figure 3. (a) Locations of the 45 sites deployed as part of the AERONET-DRAGON campaign in Washington, D.C./Baltimore metropolitan area; (b) Locations of the 25 sites deployed in South Korea during DRAGON-Asia 2012; (c) Locations of the 18 sites deployed in the San Joaquin Valley in California in late 2012 and early 2013; (d) Location of the 14 sites deployed in Japan during DRAGON-Asia 2012.

[Figure]

Figure 4. (a) Comparison of MISR V22 17.6 km resolution AODs against AERONET-DRAGON interpolated to the MISR wavelength for all cases shown in Table 2. (b) Comparison of the 4.4 km resolution AODs against AERONET-DRAGON.

[Figure]

[Figure]

Michael J Garay 3/10/2017 7:36 PM

Figure 5. (a) MISR red band image of the Osaka, Japan on 27 April 2012 at around 01:55 UTC; (b) MISR V22 17.6 km aerosol optical depth (AOD); (c) MISR 4.4 km AOD retrieved using a prototype algorithm that takes the 1.1 km resolution global data as input.

[Figure]

[Figure]

Michael J Garay 3/10/2017 7:36 PM

Figure 6. (a) MISR red band image of the Korean peninsula on 09 May 2012 at around 02:20
UTC; (b) MISR V22 17.6 km aerosol optical depth (AOD); (c) MISR 4.4 km AOD retrieved
using the prototype algorithm.

[Figure]

Figure 7. (a) MISR red band image of the Korean peninsula on 25 May 2012 at around 02:20 UTC; (b) MISR V22 17.6 km aerosol optical depth (AOD); (c) MISR 4.4 km AOD retrieved using the prototype algorithm.

Michael J Garay 3/10/2017 7:37 PM